# Remdesivir Decreases Mortality in COVID-19 Patients with Active Malignancy

**DOI:** 10.3390/cancers14194720

**Published:** 2022-09-28

**Authors:** Jerzy Jaroszewicz, Justyna Kowalska, Małgorzata Pawłowska, Magdalena Rogalska, Dorota Zarębska-Michaluk, Marta Rorat, Beata Lorenc, Piotr Czupryna, Katarzyna Sikorska, Anna Piekarska, Anna Dworzańska, Izabela Zaleska, Włodzimierz Mazur, Dorota Kozielewicz, Krzysztof Kłos, Regina Podlasin, Grzegorz Angielski, Barbara Oczko-Grzesik, Magdalena Figlerowicz, Bartosz Szetela, Beata Bolewska, Paulina Frańczak-Chmura, Robert Flisiak, Krzysztof Tomasiewicz

**Affiliations:** 1Department of Infectious Diseases and Hepatology, Medical University of Silesia in Katowice, 41-902 Bytom, Poland; 2Department of Adult’s Infectious Diseases, Medical University of Warsaw, Hospital for Infectious Diseases, 02-091 Warsaw, Poland; 3Department of Infectious Diseases and Hepatology, Faculty of Medicine, Collegium Medicum in Bydgoszcz, Nicolaus Copernicus University, 87-100 Torun, Poland; 4Department of Infectious Diseases and Hepatology, Medical University of Białystok, 15-809 Bialystok, Poland; 5Department of Infectious Diseases, Jan Kochanowski University, 25-317 Kielce, Poland; 6Department of Forensic Medicine, Wrocław Medical University, 50-367 Wroclaw, Poland; 7Pomeranian Center of Infectious Diseases, Department of Infectious Diseases, 80-210 Gdansk, Poland; 8Department of Infectious Diseases and Neuroinfections, Medical University of Białystok, 15-809 Bialystok, Poland; 9Division of Tropical and Parasitic Diseases, Faculty of Health Sciences, Medical University of Gdańsk, 80-210 Gdansk, Poland; 10Department of Infectious Diseases and Hepatology, Medical University of Łódź, 90-549 Lodz, Poland; 11Department of Infectious Diseases and Hepatology, Medical University of Lublin, 20-059 Lublin, Poland; 12Department of Paediatrics and Infectious Diseases, Wroclaw Medical University, 50-367 Wroclaw, Poland; 13Clinical Department of Infectious Diseases in Chorzów, Medical University of Silesia, 41-500 Katowice, Poland; 14Department of Infectious Diseases and Allergology, Military Institute of Medicine, 04-141 Warsaw, Poland; 15Hospital for Infectious Diseases, The IVth Department, 01-201 Warsaw, Poland; 167th Navy Hospital, 80-305 Gdansk, Poland; 17Department of Infectious Diseases and Child Neurology, Poznan University of Medical Sciences, 60-572 Poznan, Poland; 18Department of Infectious Diseases, Liver Diseases and Acquired Immune Deficiencies, Wroclaw Medical University, 50-367 Wroclaw, Poland; 19Department of Infectious Diseases, Poznań University of Medical Sciences, 61-701 Poznan, Poland; 20Department of Children’s Infectious Diseases, Provincial Jan Boży Hospital, 20-089 Lublin, Poland

**Keywords:** COVID-19, malignancy, cancer, mortality, remdesivir, real-world data

## Abstract

**Simple Summary:**

Patients with active malignancies have an increased risk for severe SARS-CoV-2 infection and high mortality from COVID-19. Additionally, due to the underlying immune deficiency, prolonged replication and a higher rate of escape mutations are seen. Thus, it is crucial to introduce direct antiviral agents, whereas there is only limited knowledge of their efficacy and optimal regimens mainly from small series in oncologic patients. In this real-world experience study, 252 patients with active malignancy were found among 4890 hospitalized patients for COVID-19. We have shown that patients with malignancy benefit from early remdesivir therapy, resulting in a decrease in 28-day in-hospital mortality by 80%. Factors independently associated with a worse prognosis include low glomerular filtration rate and low peripheral oxygen saturation at baseline. The results not only confirm the lifesaving effect of remdesivir in oncologic patients, but also underline the need to optimize therapy, including kidney protection and early oxygen therapy.

**Abstract:**

Data on the use of remdesivir, the first antiviral agent against SARS-CoV-2, are limited in oncologic patients. We aimed to analyze contributing factors for mortality in patients with malignancies in the real-world CSOVID-19 study. In total, 222 patients with active oncological disorders were selected from a nationwide COVID-19 study of 4890 subjects. The main endpoint of the current study was the 28-day in-hospital mortality. Approximately half of the patients were male, and the majority had multimorbidity (69.8%), with a median age of 70 years. Baseline SpO_2_ < 85% was observed in 25%. Overall, 59 (26.6%) patients died before day 28 of hospitalization: 29% due to hematological, and 20% due to other forms of cancers. The only factor increasing the odds of death in the multivariable model was eGFR < 60 mL/min/m^2^ (4.621, *p* = 0.02), whereas SpO_2_ decreased the odds of death at baseline (0.479 per 5%, *p* = 0.002) and the use of remdesivir (0.425, *p* = 0.03). This study shows that patients with COVID-19 and malignancy benefit from early remdesivir therapy, resulting in a decrease in early mortality by 80%. The prognosis was worsened by low glomerular filtration rate and low peripheral oxygen saturation at baseline underlying the role of kidney protection and early hospitalization.

## 1. Introduction

Since the beginning of the COVID-19 pandemic declared by the WHO on 11 March 2020, more than 450 million cases of SARS-CoV-2 infection and more than 6 million deaths due to it have been registered worldwide. Certain groups of patients with COVID-19, with an increased risk of disease progression and mortality, have been determined. In the opinion of most researchers, patients with cancer are highly vulnerable to SARS-CoV-2 infection due to frequent contact with the healthcare system, an immunocompromised state from cancer and its therapy, older age, and comorbidities, partially caused by anti-tumor treatment [1,2]. According to various studies, patients with serious malignancies showing COVID-19 symptoms have a high probability of early mortality [3].

Several pharmacological approaches have been suggested for the treatment of COVID-19. Some therapeutic approaches for COVID-19 concentrated on the repurposing of the already approved drugs, which have been known to have activity against other viruses. They included remdesivir (RDV), which had been earlier studied for the treatment of the Ebola virus, as well as SARS-CoV-1 and Middle East respiratory syndrome (MERS) coronaviruses [4,5]. Based on findings from phase III clinical trials and real-world experience studies, RDV received both American and European authorization [6,7,8].

RDV appears to be less prone to significant drug interactions than the other drugs proposed for COVID-19 treatment [9]. It also has a good profile of tolerance and shows few drug interactions, all of which make RDV a suitable drug for use in humans. Nevertheless, when the efficacy of RDV was evaluated in RCTs, the real advantage of RDV was questioned. A meta-analysis, including five RCTs studying over 7000 COVID-19 patients failed to demonstrate that RDV administration was associated with an improvement in survival. Other meta-analyses pointed out that RDV improved clinical outcomes but did not reduce mortality, in comparison to untreated patients [10,11,12]. These conflicting results are attributable to many factors (i.e., steroid use, age, immunosuppression, and timing of administration), which can influence the effects of antiviral treatments, mostly because of the difficulties in the selection of optimal patient populations and finding the appropriate stage of the disease for remdesivir administration.

IDSA recommends remdesivir for treatment of severe COVID-19 in hospitalized patients with SpO_2_ (oxygen peripheral blood saturation) < 94% on room air. However, the guideline panel’s recommendation is against the routine initiation of remdesivir among patients on invasive ventilation and/or ECMO (extracorporeal membrane oxygenation). Immunocompromised patients who are unable to control viral replication may still benefit from remdesivir despite SpO_2_ that exceeds 94% on room air or a requirement for mechanical ventilation [13]. The literature describes cases of successful use of remdesivir also in combination with high-titer convalescent plasma or with monoclonal antibodies in immunocompromised patients [14,15]. Patients with malignancies in the severely immunocompromised group might benefit from RDV therapy outside of the commonly accepted framework. Therefore, we aimed to analyze contributing factors for mortality in a large real-world COVID-19 study with a special focus on the effect of antiviral therapy.

## 2. Materials and Methods

Patients who were analyzed in the current study originated from the SARSTer national database, which included 4890 COVID-19 subjects treated between 1 March 2020 and 30 June 2021 in 30 Polish centers. SARSTer is an ongoing national real-world experience study assessing treatment outcomes in patients with COVID-19. This project is supported by the Polish Association of Epidemiologists and Infectiologists (PTEiLChZ) and the Medical Research Agency.

All the patients were diagnosed with SARS-CoV-2 based on positive results of the real-time reverse transcriptase–polymerase chain reaction (RT-PCR) from the nasopharyngeal swab specimen and required hospitalization due to COVID-19. The decision concerning the treatment regimen was taken entirely by the treating physician based on current knowledge and national recommendations [16,17]. If selected remdesivir was administered intravenously once daily for 5 days, with a loading dose of 200 mg on day 1, followed by a maintenance dose of 100 mg; tocilizumab also iv in a single dose of 800 mg if PBW > 90 kg; 600 mg if PBW > 65 kg and ≤90 kg; 400 mg if PBW > 40 kg and ≤65 kg; and dexamethasone usually orally or intravenously with doses 4–8 mg per day.

The current study included 255 adults (≥18 yrs) patients with active oncological disorders, who constituted 5.2% of subjects included in the SARSTer database (N = 4890) at the time of analyses. Patients with oncological disease indicated in their medical history or who were in remission (n = 24), and without information on COVID-19 outcome (n = 9), were excluded from the study. In total, 222 patients underwent further analyses. Data were entered retrospectively and submitted online by a web-based platform operated by Tiba sp. z o.o. Baseline data included age, gender, body mass index (BMI), comorbidities, and other medications.

COVID-19, SpO_2_, and laboratory measures of inflammation, including C-reactive protein (CRP), procalcitonin, white blood cells (WBC), platelets, interleukin 6 (IL-6), and d-dimers. In the current study as an endpoint of treatment, efficacy served early hospital mortality before day 28 of hospitalization. Multimorbidity was defined as having at least one of the following diagnoses in addition to ongoing malignancy: diabetes mellitus, cardiovascular disease, and pulmonary disease.

The results were expressed as n (%) or median (interquartile range, IQR). In statistical analyses groups were compared with non-parametric tests, Fisher’s exact test for categorical and the Kruskal–Wallis test for continuous variables. All tests of significance were two-sided. Logistic regression models were used to identify factors associated with the odds of early hospital death. Factors significant in univariate models (*p* < 0.1) were included in the final multivariate model. A confidence interval (CI) of 95% was accepted. For continuous variables, odds ratios (OR) in uni- and multivariate analyses were shown per incremental unit of the parameter from the median. All analyses were performed using SAS version 9.4 (SAS Institute, Cary, NC, USA).

The study was conducted according to the guidelines of the Declaration of Helsinki, and approved by the Ethics Committee of the Medical University of Białystok (29 October 2020, number APK.002.303.2020). Additionally, the local bioethics committees approved the experimental use of tocilizumab in patients with COVID-19, and written informed consent was obtained from each patient before the treatment initiation.

## 3. Results

Of 222 patients included in the analyses, 60 (27.0%) had hematological and 162 (73.0%) other types of malignancy. Approximately half of the patients were male (51.8%), the majority had multimorbidities (69.8%), the median age was 70 (IQR: 63.0–78.0) years, and the median BMI was 26.5 (23.5–29.8) kg/m^2^. Baseline laboratory values are presented in Table 1. In terms of disease severity 25% of patients had SpO_2_ below 85% (median 92% IQR: 85–95%). In terms of disease progression, 59 (26.6%) of patients died before or on day 28 of hospitalization, 47 (29%) in the group with hematological cancer, and 12 (20%) in the group with other cancers (*p* = 0.231).

Since gender is a well-known factor affecting mortality in patients with malignancy we also compared baseline characteristics for both genders (Appendix A). Interestingly enough, woman who died of cancer were older than those who survived (73 (68–81) vs. 69 (62–78) years old, *p* = 0.09), whereas such a trend was not observed in men (69 (63–81) vs. 70 (65–76) years old, *p* = 0.54). This might further suggest an additional influence of gender, irrespective of age, in COVID-19 patients.

Fifty-eight (26.1%) patients received remdesivir, in the majority of cases for five days with a median time from symptoms onset of 6 days and a median time since diagnosis of 5 days. One hundred and ten (49.5%) patients received dexamethasone, with a median time of therapy duration of 9 days and a median time of treatment initiation since diagnosis of 2 days. Additionally, thirty-six (16.2%) patients were treated with tocilizumab, and forty-three (19.4%) were treated with convalescent plasma (Table 1).

Patients with hematological cancers were comparable to patients with other types of cancer in respect to gender, age, BMI, and SpO_2_ at hospital admission, as well as in most other characteristics. However, patients with hematological malignancies were significantly more likely to have low neutrophils (2700 vs. 4180/μL, *p* = 0.0002) and platelet (1042 vs. 192.000/μL, *p* < 0.0001) counts and less likely to have multimorbidity (56.7% vs. 74.7%, *p* = 0.0132). In terms of COVID-19-specific treatment, patients with hematological cancers were more likely to receive convalescent plasma (33.3% vs. 14.2%, *p* = 0.0021), as well as to receive dexamethasone and tocilizumab later after diagnosis of COVID-19 (3 vs. 2 (IQR: 1–7 vs. 1–5) days and 4 vs. 4 (9.5–15 vs. 2–7) days, *p* = 0.0423 and *p* = 0.0425, respectively), Table 2.

Interestingly enough, the baseline characteristics of patients who died were comparable to those who survived in terms of age, BMI, type of cancer, multimorbidity, and use of other medication (Table 1). However, patients who died were significantly more likely to be male (66.1% vs. 46.6%, *p* = 0.0146), to have lower SpO_2_ at hospital admission (84% vs. 92%, *p* < 0.0001), lower lymphocyte count (670 vs. 990/μL, *p* = 0.0014), more often eGFR < 60 mL/min/m^2^ (31% vs. 45%, *p* = 0.0004) and higher CRP (96.5 vs. 57.3 mg/dL, *p* = 0.0008), procalcitonin (0.27 vs. 0.11 ng/mL, *p* = 0.0017), IL-6 (139.7 vs. 43.9 pg/mL, *p* < 0.0001), neutrophils (5190 vs. 3260/μL, *p* = 0.0091).

In terms of COVID-19-specific treatment patients who died were less likely to receive remdesivir (15.5% vs. 30.1%, *p* = 0.0370); however, there was no difference in the time of starting treatment from the onset of symptoms or diagnosis. Those who died were also more likely to receive dexamethasone (62.7% vs. 44.8%, *p* = 0.0226) with a shorter time of treatment duration (8 vs. 9 days, *p* = 0.0233), but with no difference in the time since diagnosis. There were no statistical differences in terms of using tocilizumab and convalescent plasma between those who died and those who survived, Table 1.

In univariate logistic regression model factors associated with increased odds of early hospital death were male sex (OR 2.174 [95%CI:1.200–3.939], *p* = 0.0104), older age (1.309 [1.022–1.667] per 10 years increase, *p* = 0.0330) and eGFR < 60 mL/min/m^2^ (3.055 [1.671–5.584], *p* = 0.0003). In terms of COVID-19-specific treatment, the only factors associated with the likelihood of death were remdesivir, which decreased (0.425 [0.201–0.895], *p* = 0.0243) the odds, and dexamethasone, which increased (2.121 [1.175–3.827], *p* = 0.0125) the odds. In addition, several laboratory values were associated with the odds of death including CRP (1.272 [1.082–1.496] per 50 mg/dL increase, *p* = 0.0036). procalcitonin (1.210 (1.019–1.435] per 2 ng/mL increase, *p* = 0.0292), neutrophil count (1.120 [1.050–1.195], per 1000/μL decrease, *p* = 0.8045), IL-6 (1.239 [1.059–1.450] per 200 pg/mL increase, *p* = 0.0074) and d-dimers (1.159 [1.025–1.309] per 2000 ug/mL increase, *p* = 0.0185) (Table 3).

The final multivariate model included 126 patients and 22 deaths. After including all significant factors from the univariable model, the only factor increasing the odds of death in the multivariable model was eGFR < 60 mL/min/m^2^ (4.621 [1.244–17.17], *p* = 0.0223). The only independent factor significantly decreasing the odds of death among patients with malignancy and COVID-19 was SpO_2_ (0.479 [0.303–0758] per 5% increase, *p* = 0.0017) and the use of remdesivir (0.425 [0.201–0.895], *p* = 0.0314), Figure 1.

## 4. Discussion

In this cohort of patients with various malignancies, including almost one-third with hematologic cancer in the univariate analysis, we found male sex and age to be risk factors for early hospital death. Similar observations were made in a number of analyses. In a large cohort analysis published by Lee et al. and Williams et al., the risk of death was significantly associated with age and male sex [18,19]. On the other hand, when we performed a multivariate logistic regression model, both sex and age lost their significance. Similar results were observed in cohorts of patients with COVID-19 and thoracic cancers. The TERAVOLT registry included patients from eight countries, and in univariate analyses identified age > 65, current or former smokers, presence of any comorbidities, and chemotherapy as significant risk-of-death factors; however, multivariate analysis only revealed smoking history as associated with an increased risk of death [20]. One may conclude that the type of statistical analysis, especially the need for multifactorial analyses, may have a great impact on final conclusions. In this case, the lack of influence of age may result from the fact that most of the patients in our cohort were already at high risk, with all having baseline active malignancy, most being over 60 years old.

SpO_2_ at hospital admission and decreased remained crucial factors associated with odds of death. These data do not require an explanation as it reflects the severity of the disease course prior to hospitalization and/or preexisting comorbidities, particularly kidney insufficiency and respiratory failure. Both are well known and described in multiple studies. For example, Mamlouk O et al. found the overall mortality rate of cancer patients with CKD and COVID-19 was almost 1.7 times higher than the rate of patients with no CKD at the tertiary cancer center in Houston [21]. Additionally, in one of the studies from the SARSTER project, we were able to find that lower eGFR was a leading driving factor of mortality in the analysis of 2322 COVID-19 patients where a subject with eGFR < 30 mL/min had six-times higher mortality compared with those >60 mL/min [22].

We did not find multimorbidity as an independent predictor of early hospital death, which may reflect that most of the patients in our cohort were already at high risk with all having as baseline disease active cancer. In addition, BMI was not identified as a risk factor; however, we believe that in cancer patients, who might be underweight, the increase in BMI reflects an improvement in underlying disease rather than deterioration.

In most analyses, the type of cancer might be a critical prognostic factor for both course severity and final outcome. Retrospective studies suggest that higher rates of case fatality are observed in patients with hematologic malignancies than in those with solid tumors [23]. Our study group consisted of patients with different types of cancer. Importantly, we could not find a difference in 28-day mortality between the subjects with hematologic vs. other types of cancers, most likely due to the relatively low number of patients. Importantly, however, groups of patients with hematological cancer and other types of cancer were comparable with respect to gender, age, BMI, and, most importantly, SpO_2_ at hospital admission. Furthermore, we were not able to analyze separately prognostic factors of death among patients with specific types of cancers.

In our analyses, the use of remdesivir was the only independent factor significantly decreasing the odds of early hospital death by over 80% among patients with malignancy. For the study group, irrespective of the type of cancer, the median number of days from symptom onset for introducing antiviral remdesivir treatment was 6 days. After comparison of those who died with patients who survived, the initiation of remdesivir treatment was slightly delayed, although there was no significant difference. This may indicate that prompt and timely initiation of antiviral therapy may have a beneficial effect in some subgroups of patients, such as those with immunosuppression. Importunately, for the number of hematologic disorders, prolonged viral shedding (detection of SARS-CoV-2 by molecular testing) may occur, although it how it impacts hospital deaths remains unknown [24].

According to general and focused cancer patient guidelines, remdesivir is recommended for inpatients hospitalized for COVID-19 pneumonia, with hypoxia, but who are not yet requiring mechanical ventilation or ECMO [25]. In those hospitalized with COVID-19 pneumonia, patients receiving remdesivir had a shorter time to recovery, clinical improvement, and lower mortality [6]. A study of patients without hypoxia (with oxygen saturation >94% on room air) found inconsistent results, with a 5-day course of therapy with remdesivir resulting in improved clinical status but no impact with longer courses. Patients with COVID-19 without reduced oxygen saturation included in the SIMPLE-moderate phase III trial treated with remdesivir for 5 days improved clinically when compared with those receiving standard of care alone. Similar results were obtained for patients with the severe form of COVID-19 defined as oxygen saturation SpO_2_ < 95% or receiving oxygen support, treated with remdesivir in the SIMPLE-severe phase III trial [26,27].

The ACTT-1 and SIMPLE trials also showed that the safety profile of remdesivir was similar to that of placebo. Although some open-label trials involving hospitalized patients showed conflicting results of clinical efficacy, refs. [26,28,29] data from the ACTT-1 and SIMPLE trials and multiple real-world studies, including previous SARSTer analysis [8], together with the results of the current trial, proved the clinical benefit of treatment with remdesivir in different populations of COVID-19 patients.

In none of the previously published studies were oncology patients analyzed. There is a number of case studies with single or a series of cases discussing rationales for the use and results of remdesivir treatment in cancer patients. Hence, our study is one of the first oriented at oncologic patients.

Surprising results were shown with respect to the use of dexamethasone. In univariate analysis, this treatment was associated with worse prognosis and higher early hospitalization death in cancer patients (*p* = 0.0243), whereas in multivariate this significance was lost (*p* = 0.2824). It is contrary to most of the recommendations and a number of studies. A meta-analysis of seven clinical trials representing 1703 critically ill patients found decreased mortality when different corticosteroids were used for COVID-19 treatment; nevertheless, the effect only reached statistical significance with dexamethasone [30]. Some important observations were reported in the RECOVERY trial. Corticosteroids were found to be harmful in patients who did not require supplemental oxygen support at baseline [31]. A clinical trial of 299 hospitalized patients with moderate-to-severe COVID-19 reported improved outcomes (less need for mechanical ventilation and better Sequential Organ Failure Assessment [SOFA] scores), but no significant impact on mortality [32]. The beneficial effect of dexamethasone was reported for severe and/or late COVID-19 patients, whereas in our study, dexamethasone was introduced relatively early, during the first 5 days from diagnosis (median 2 days). We may conclude that early dexamethasone treatment showed no benefit to the final disease outcome.

As mentioned, our study is one of the first to analyze treatment effectiveness in cancer patients. We are aware of the limitations of our study. Among them, the impact of multiple factors related to cancer and COVID-19 treatment should be pointed out. One of the major confounding factors was the type of cancer and oncologic treatment. One should mention also that comedications of COVID-19 and their impact on the patient’s condition may influence the course of the disease and potential complications. To eliminate this imbalance as a confounding factor, we performed an analysis using multivariate logistic regression. On the other hand, the major strength of the study lies in the collection of data from a real-world, heterogeneous population, thus being representative of routine practice. We believe that our observations may help to improve our understanding of the complexity of COVID-19 cancer patient management. As SARSTer is an ongoing nationwide study, we will re-evaluate these observations with increasing numbers of patients in the cohort.

## 5. Conclusions

In conclusion, this real-world experience study shows that patients with malignancies could be among one of the populations to especially benefit from early remdesivir therapy, with a decrease in 28-day in-hospital mortality of 80%. On the other hand, leading factors independently associated with worse prognosis in patients with cancers include low glomerular filtration rate and low peripheral oxygen saturation, which underline the role of kidney protection and early hospitalization in this subgroup of COVID-19 patients.

## Figures and Tables

**Figure 1 cancers-14-04720-f001:**
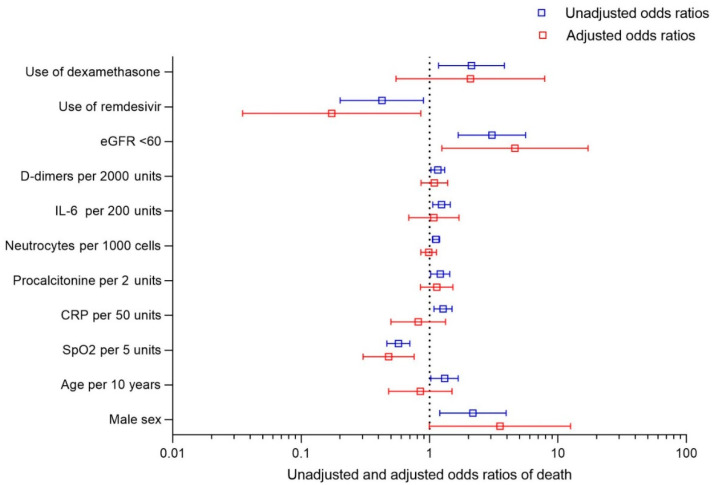
Unadjusted and adjusted odds ratio of 28-day mortality in COVID-19 patients with active malignancy.

**Table 1 cancers-14-04720-t001:** Baseline characteristics of studied cohort stratified by early hospital death.

Characteristic	Nr with Data	All N = 222	Died N = 59	Survived N = 163	*p* Value
Male sex, n (%)	222	115 (51.8)	39 (66.1)	76 (46.6)	0.0146 *
Age in years, median (IQR)	222	70.0 (63.0–78.0)	70 (63–81)	70 (63–78)	0.3105 *
BMI in kg/m^2^, median (IQR)	175	26.5 (23.5–29.8)	26.0 (23.9–31.0)	26.5 (23.5–29.4)	0.7718 **
Hematological cancer, n (%)	222	60 (27.0)	12 (29.3)	48 (29.4)	0.2310 *
SpO_2_ in % at hospital admission, median (IQR)	221	92 (85–95)	84 (78–94)	92 (88–95)	<0.0001 **
CRP baseline [mg/dL], median (IQR)	222	88.3 (26.2–134.7)	96.5 (40.7–155)	57.3 (22–113.3)	0.0008 **
Procalcitonin baseline [ng/mL], median (IQR)	167	0.13 (0.05–0.37)	0.27 (0.1–0.91)	0.11 (0.05–0.3)	0.0017 **
WBC baseline [/μL], median (IQR)	221	5500 (3700–9150)	7840 (3970–12,860)	5360 (3610–8060)	0.1280 **
Lymphocytes baseline [/μL], median (IQR)	214	895 (590–1300)	670 (490–950)	990 (660–1350)	0.0014 **
Neutrocytes baseline [/μL], median (IQR)	214	3855 (2290–6930)	5190 (2380–10,460)	3260 (2290–5690)	0.0091 **
PLT baseline [/μL], median (IQR)	221	178,000 (127,000–255,000)	164,000 (117,000–251,000)	178,500 (134,000–255,000)	0.5771 **
IL-6 baseline [pg/mL], median (IQR)	132	49.1 (22.8–116.8)	139.7 (52.7–344.3)	43.9 (17.4–90.5)	<0.0001 **
D-dimers baseline [μg/mL], median (IQR)	202	1228 (755–2290)	1712 (1041–4152)	1095 (699–1929)	0.0025 **
eGFR < 60 mL/min/m^2^ baseline, n (%)	220	76 (34.5)	31 (54.4)	45 (27.6)	0.0004 *
Use of other medication at baseline, n (%)	219	188 (85.8)	51 (87.9)	137 (85.1)	0.6666 *
Multimorbidity	222	155 (69.8)	112 (68.7)	43 (72.9)	0.6212 *
Use of RDV, n (%)	-	58 (26.1)	9 (15.5)	49 (30.1)	0.0370 *
Days from symptoms, median (IQR):	66	6 (4–8)	8 (6–9)	6 (4–8)	0.1279 **
Days from diagnosis, median (IQR):	68	2 (1–5)	1.5 (1–5)	2 (2–4.5)	0.4186 **
Days on RDV, median (IQR):	68	5 (5–5)	5 (4.5–5)	5 (5–5)	0.0047 **
Use of TCZ, n (%)	-	36 (16.2)	13 (22.0)	23 (14.1)	0.2150 *
Days from symptoms, median (IQR):	34	10 (8–13)	9 (8–10)	10 (7–14)	0.7048 **
Days from diagnosis, median (IQR):	38	5 (2–9)	4 (3–8)	5 (2–10)	0.9736 **
Use of dexamethasone, n (%)	-	110 (49.5)	37 (62.7)	73 (44.8)	0.0226 *
Days from diagnosis, median (IQR):	125	2 (1–5)	1 (1–5)	3 (1–5)	0.1184 **
Days on dexamethasone, median (IQR):	123	9 (6–12)	8 (3–12)	9 (7–12)	0.0233 **
Use of convalescent plasma, n (%)	-	43 (19.4)	8 (13.6)	35 (21.5)	0.2488 *
Days from diagnosis, median (IQR):	52	4 (2–8)	3 (2–7)	4 (2–8)	0.9249 **

* Fisher exact test. ** Kruskal–Wallis test; Abbreviations: BMI—body mass index; IQR—interquartile range; SpO_2_—oxygen peripheral blood saturation; CRP—C-reactive protein; WBC—white blood cells, PLT—platelets; RDV—remdesivir; TCZ—tocilizumab.

**Table 2 cancers-14-04720-t002:** Baseline characteristics of studied cohort stratified by the type of cancer.

Characteristic	Nr with Data	All N = 222	Other N = 162	Hematological N = 60	*p* Value
Male sex, n (%)	222	115 (51.8)	90 (55.6)	25 (41.7)	0.0713 *
Age in years, median (IQR)	222	70.0 (63.0–78.0)	70 (64.0–79.0)	69.0 (57.0–78.0)	0.1550 **
BMI in kg/m^2^, median (IQR)	175	26.5 (23.5–29.8)	27.7 (23.9–30.5)	25.3 (23.0–28.6)	0.0703 **
SpO_2_ in % at hospital admission, median (IQR)	221	92 (85–95)	91 (85–95)	92 (89–95)	0.5531 **
CRP baseline [mg/dL], median (IQR)	222	88.3 (26.2–134.7)	68.3 (24.0–133.0)	58.3 (31.1–141.6)	0.3484
Procalcitonin baseline [ng/mL], median (IQR)	167	0.13 (0.05–0.37)	0.13 (0.05–0.31)	0.13 (0.06–0.45)	0.5876 **
WBC baseline [/μL], median (IQR)	221	5500 (3700–9150)	6070 (3970–9250)	4795 (2750–7685)	0.0368 **
Lymphocytes baseline [/μL], median (IQR)	214	895 (590–1300)	900 (640–1330)	900 (390–1300)	0.1504 **
Neutrocytes baseline [/μL], median (IQR)	214	3855 (2290–6930)	4180 (2500–7300)	2700 (1090–4360)	0.0002 **
PLT baseline [/μL], median (IQR)	221	178,000 (127,000–255,000)	192,000 (146,000–266,000)	1042 (755–1741)	<0.0001 **
IL–6 baseline [pg/mL], median (IQR)	132	49.1 (22.8–116.8)	50.1 (18.1–114.2)	45.7 (28.8–107.2)	0.4065 **
D–dimers baseline [μg/mL], median (IQR)	202	1228 (755–2290)	1301 (743–2590)	1042 (755–1742)	0.3318 **
eGFR < 60 mL/min/m^2^ baseline, n (%)	220	76 (34.5)	105 (65.6)	39 (65.0)	1.0000 *
Use of other medication at baseline, n (%)	219	188 (85.8)	134 (83.7)	54 (91.5)	0.1903 *
Multimorbidity	222	155 (69.8)	121 (74.7)	34 (56.7)	0.0132 *
Died, n (%)	222	59 (26.6)	47 (29.0)	12 (20.0)	0.2310 *
For patients receiving RDV, median(IQR):	–	58 (26.1)	39 (24.1)	19 (31.7)	0.3021 *
Days from symptoms, median(IQR):	66	6 (4–8)	7 (5–9)	5 (3–8)	0.0832 **
Days from diagnosis, median(IQR):	68	2 (1–5)	2 (2–5)	2 (1–5)	0.2000 **
Days on RDV, median(IQR):	68	5 (5–5)	5 (5–5)	5 (5–5)	0.1028 **
Use of TCZ, n (%)	–	36 (16.2)	28 (17.3)	8 (13.3)	0.5441 *
Days from symptoms, median(IQR):	34	10 (8–13)	9 (5–12)	12.5 (10–15)	0.1147 **
Days from diagnosis, median(IQR):	38	5 (2–9)	4 (2–7)	4 (9.5–15)	0.0425 **
Use of dexamethasone, n (%)	–	110 (49.5)	77 (47.5)	33 (55.0)	0.3657 *
Days from diagnosis, median(IQR):	125	2 (1–5)	2 (1–5)	3 (1–7)	0.0423 **
Days on dexamethasone, median(IQR):	123	9 (6–12)	9 (6–12)	8 (6–10)	0.7506 **
Use of convalescent plasma, n (%)	–	43 (19.4)	23 (14.2)	20 (33.3)	0.0021 *
Days from diagnosis, median(IQR):	52	4 (2–8)	4 (2–7)	6 (2–14)	0.5002 **

* Fisher exact test. ** Kruskal–Wallis test; Abbreviations: BMI—body mass index; IQR—interquartile range; SpO_2_—oxygen peripheral blood saturation; CRP—C-reactive protein; WBC—white blood cells, PLT—platelets; RDV—remdesivir; TCZ—tocilizumab.

**Table 3 cancers-14-04720-t003:** Univariate and multivariate logistic regression models for the odds of death.

	Univariate		Multivariate *	
Characteristic	OR (95% CI)	*p* Value	OR (95% CI)	*p* Value
Male sex	2.174 (1.200–3.939)	0.0104	3.529 (0.994–12.53)	0.0511
Age per 10 years	1.309(1.200–3.939)	0.0330	0.847 (0.479–1.496)	0.5667
BMI per 1 unit	1.008 (0.940–1.081)	0.8261	-	
SpO_2_ at hospital admission per 5 units [%]	0.571 (0.465–0.701)	<0.0001	0.479 (0.303–0.758)	0.0017
CRP per 50 units [mg/dL]	1.272 (1.082–1.496)	0.0036	0.816 (0.500–1.331)	0.4154
Procalcitonin per 2 units [ng/mL]	1.210 (1.019–1.435)	0.0292	1.136 (0.850–1.518)	0.3888
WBC per 1000 units [/μL]	1.026 (0.995–1.058)	0.1012	-	
Leukocytes per 100 units [/μL]	0.994 (0.981–1.008)	0.4278	-	
Neutrocytes per 1000 units [/μL]	1.120 (1.050–1.195)	0.0006	0.983 (0.855–1.129)	0.8045
Platelets per 10,000 [/μL]	0.994 (0.968–1.020)	0.6315	-	
IL-6 per 200 units [pg/mL]	1.239 (1.059–1.450)	0.0074	1.079 (0.688–1.693)	0.7411
D-dimers per 2000 units [μg/mL]	1.159 (1.025–1.309)	0.0185	1.088 (0.857–1.381)	0.4866
eGFR < 60 mL/min/m^2^	3.055 (1.671–5.584)	0.0003	4.621 (1.244–17.17)	0.0223
Use of other medication	1.509 (0.625–3.641)	0.3604	-	
Multimorbidity (CVD, COPD, Asthma, DM)	1.234 (0.652–2.336)	0.5188	-	
Hematological cancer	0.621 (0.306–1.261)	0.1878	-	
Use of remdesivir	0.425 (0.201–0.895)	0.0243	0.173 (0.035–0.855)	0.0314
Use of tocilizumab	1.712 (0.812–3.611)	0.1578	-	
Use of dexamethasone	2.121 (1.175–3.827)	0.0125	2.077 (0.548–7.880)	0.2824
Use of convalescent plasma	0.829 (0.394–1.744)	0.6213	-	

* Adjusted for all significant in univariable models; Abbreviations: BMI—body mass index; SpO_2_—oxygen peripheral blood saturation; CRP—C-reactive protein; WBC—white blood cells, PLT—platelets; CVD—cardiovascular disorder; COPD—chronic obstructive pulmonary disease; DM—diabetes mellitus.

## Data Availability

The data can be shared up on request.

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
