# Peer review of "Remdesivir Decreases Mortality in COVID-19 Patients with Active Malignancy"

_cancers, 2022, doi:10.3390/cancers14194720_

Round 1
Reviewer 1 Report
In this manuscript, the authors present the data on efficacy of Remdesivir in decreasing mortality in covid patients previously diagnosed with different types of cancers. The data is interesting as early use of Remdesivir can reduce up to 80% mortality rate in the patients with Covid19/cancer. Though the study is not complete per se, it presents significant insights on efficacy of antiviral drugs such as Remdesivir in cancer patients when infected with Covid19. The manuscript needs to be proof-read thoroughly for inadvertent typos and language including the abstract section, line 62.
Author Response
Thank you for your comment. The manuscript has undergone a thorough linguistic revision of both spelling and grammar. Individual changes were not marked due to their multiplicity.
Reviewer 2 Report
This is an important study on the effect of remdesivir in COVID-19 patients with active malignancy. It may be more informative if each parameters of male and female are shown in Tables if possible.
Author Response
Thank you for your comment. In accordance with the comments of the reviewer, an additional table (Supplementary Table 1) has been created which illustrates the initial parameters of the analyzed group depending on gender. A corresponding adnotation has also been added in the results section. Gender was also considered in the final multivariate analysis.
Round 2
Reviewer 1 Report
The authors have significantly improved the manuscript by editing the language and style of presentation. The manuscript may be considered for publication.